# Drug Conjugated and Bispecific Antibodies for Multiple Myeloma: Improving Immunotherapies off the Shelf

**DOI:** 10.3390/ph14010040

**Published:** 2021-01-07

**Authors:** Gregorio Barilà, Rita Rizzi, Renato Zambello, Pellegrino Musto

**Affiliations:** 1Department of Medicine (DIMED), Hematology and Clinical Immunology Section, Padova University School of Medicine, 35121 Padova, Italy; gregorio.barila@gmail.com; 2Department of Emergency and Organ Transplantation, “Aldo Moro” University School of Medicine, 70121 Bari, Italy; rita.rizzi@uniba.it; 3Unit of Hematology and Stem Cell Transplantation, AOUC Policlinico, 70124 Bari, Italy

**Keywords:** immunotherapy, antibodies drug conjugated, bispecific antibodies, multiple myeloma therapy

## Abstract

The impressive improvement of overall survival in multiple myeloma (MM) patients in the last years has been mostly related to the availability of new classes of drugs with different mechanisms of action, including proteasome inhibitors (PI), immunomodulating agents (IMiDs), and monoclonal antibodies. However, even with this increased potence of fire, MM still remains an incurable condition, due to clonal selection and evolution of neoplastic clone. This concept underlines the importance of immunotherapy as one of the most relevant tools to try to eradicate the disease. In line with this concept, active and passive immunotherapies represent the most attractive approach to this aim. Antibody-drug conjugate(s) (ADCs) and bispecific antibodies (BsAbs) include two innovative tools in order to limit neoplastic plasma cell growth or even, if used at the time of the best response, to potentially eradicate the tumoral clone. Following their promising results as single agent for advanced disease, at the recent 62nd ASH meeting, encouraging data of several combinations, particularly of ADC(s) with PI or IMiDs, have been reported, suggesting even better results for patients treated earlier. In this paper, we reviewed the characteristics, mechanism of action, and clinical data available for most relevant ADC(s) and BsAbs.

## 1. Immunotherapy in Multiple Myeloma

Immune dysregulation retains a crucial role in the pathogenesis and disease progression of multiple myeloma (MM), since genetic lesions per se are necessary but not sufficient to the progression from monoclonal gammopathy of undetermined significance (MGUS) to overt MM [1,2]. Several factors including T cell exhaustion, tolerance induction by tumor associated microenvironment, cytokines production alteration and increase in myeloid-derived suppressor cells (MDSCs), and tumor-associated macrophages with suppressive properties contribute in the neoplastic escape form immune surveillance, ultimately leading to disease progression [3,4,5,6]. Allogenic stem cell transplantation is traditionally regarded as the first immunotherapeutic strategy and the only curative option for many hematological malignancies. However, allogenic T cells generate a modest graft vs. myeloma response, thus limiting the clinical benefit for patients [7]. Moreover, graft versus host disease and transplant related mortality confine this regimen only in selected cases on clinical trials [8,9]. Consequently, in the last years, different immune-based strategies have been developed to overcome drug resistance and disrupt the tumor suppressive milieu. These approaches can be mainly resumed in three categories:Agents that reverse tumor-related immune paralysis like immunomodulatory drugs (IMIds) and checkpoint inhibitors; these strategies act as immune booster and enhance the natural anticancer immune response;Agents directly targeting neoplastic cells including monoclonal antibodies and antibodies-drug conjugates (ADCs); this strategy represents a targeted therapy acting as passive immunotherapy since they can directly kill neoplastic cells and not always require an intact patient’s cellular immunity to exert their antitumoral properties;Agents directly activating the immune system against neoplastic cells like Chimeric Antigen Receptor expressing T-cell (CAR-T), Bispecific T cell redirecting antibodies (BsAbs), and peptide-based vaccines [10].

In this review, we will focus on available, though still preliminary, clinical results of immunotherapeutic approaches specifically targeting neoplastic cells and/or immune system like ADC(s) and BsAbs.

## 2. Antibodies Drug Conjugated

The possibility to introduce a cytotoxic agent in a tumor cell upon recognition by a specific antibody represents the rationale for ADC(s) therapy. This combination of chemo and immunotherapy is likely to be enclosed in the concept of “Magic Bullets”, coined in the early 1900s by Nobel prize winner Paul Ehrlich. An ADC is composed of different parts with crucial relevance for its efficacy by sparing toxicity (Figure 1).

The first is represented by a monoclonal antibody (mAb) or fragment of mAb, which recognizes a tumor-associated antigen, giving specificity to the product. The second peculiar feature is that this mAb is armed with a cytotoxic drug (also known as payload). When the ADC binds to its target antigen at the cell surface, the antigen/ADC complex is internalized within the cell in the endo-lysosomal compartment. Here, the ADC can be disrupted by linker cleavage or antibody degradation and the payload is released and free to interfere with vital cellular functions.

A crucial step for this process to be effective is that the payload cannot be released by standing. This needs a specialized chemical linker, which binds the cytotoxic product to the antibody [11] in order to prevent accidental release.

Taken together, this process combines the selectivity of the antibody and toxicity of the payload against a specific target, limiting adverse effects. It should also be pointed out that, depending on the immune status of the patient, this process is also combined with Fc dependent function, including antibody dependent cell toxicity (ADCC) and antibody dependent cell phagocytosis (ADCP) [12] (Figure 2).

Considering the mechanism of action, several conditions must be met in order to confer best efficacy and safety to ADC(s). Careful selection of antigen target represents a crucial point. Optimally, it should be highly expressed on neoplastic cells whereas lacking in normal tissues, to minimize off target uptake. This condition, however, can rarely be satisfied because of heterogeneous antigen expression between tumors as well as patients, the rate of internalization of antibody, and antigen complex conditioning efficacy [11]. Due to its longer half-life, human IgG1 is most often chosen: in addition, this IgG subclass mediated most of ADCC and Complement Dependent Cytotoxicity (CDC) [13]. Finally, to increase binding affinity to Fc domain, antibodies can be afucosilated [14].

The payloads used in ADC(s) are selected small molecules with high potency and proper hydrophobicity. The payloads commonly used in ADC(s) account for anti-mitotic agents since non-malignant cells have lower mitotic rates [11] and can be divided into two main categories: microtubule inhibitors and DNA-damaging agents. To the first category belong auristatins and maytansinoids, which bind to tubulin, causing G2/M arrest and apoptosis [13]. Auristatins include monomethyl auristatin E (MMAE, vedotin such as brentuximab vedotin and polatuzumab vedotin) and monomethyl auristatin F (MMAF, mafodotin) [15]. Relevant difference between the two molecules is the possibility to exert a bystander effect to neighboring tumour cells by MMAE due to its membrane permeability, while MMAF lacks this ability. DNA damaging agents are other molecules that can be used, causing cell death by the induction of DNA double-strand breaks [15]. An example of this class of payload is calicheamicin, used in inotuzumab ozogamicin and gemtuzumab ozogamicin.

In order to prevent off target toxicity, an ideal linker should not prematurely release the payload. Linkers currently used in ADC(s) fall into two broad categories: cleavable and non-cleavable linkers. Cleavable linkers are sensitive to different intracellular conditions: pH, lysosomal protease or glutathione [16,17]. These linkers are less stable than non-cleavable linkers, leading the possibility of off target side-effects. Non cleavable linkers rely on complete proteolytic degradation of the whole ADC(s) complex by the lysosomes to release active payloads. The optimal ratio between payload and antibody is usually 3 to 4:1 [11]. This ratio strongly influences drug stability, thus impacting the potential toxic effect of the complex. As a matter of fact, toxicity can be the result of different conditions, namely expression of target antigen on normal tissue, inadequate linker stability leading to systemic release of the payload, or non-specific/off-target uptake of the linker-payload compound [18]. In particular, MMAE (as the case of Brentuximab) has been mostly associated with peripheral neuropathy, while MMAE and MMAF are associated with hematological toxicity when attached by non-cleavable linkers [15]. Ocular toxicity represents another distinctive adverse event of these payloads. Hepatic toxicity and thrombocytopenia have been associated with calicheamicin payloads [18].

## 3. Antigen Target of ADC(s)

As stated above, the selection of a suitable antigen is crucial for efficacy of the compound. Optimally, it should be highly expressed on neoplastic cells while lacking in normal tissues, to possess internalization capability in order to introduce the complex within the cell and to maintain stably expression during time even in the presence of specific target antibody. Although any target fully includes all these features, B cell maturation antigen (BCMA) has emerged as a very promising target [17]. Other targets which have been considered are represented by CD138 (syndecan) [19], CD56 [20], CD38 [21], CD74 [22], and CD46 [23]; however, due to unsatisfying results, the clinical development of ADC(s) targeting CD138 and CD56 has been halted or different approaches are being used (immunotoxin as for anti-CD38) and they will not be discussed in this review.

### 3.1. B Cell Maturation Antigen (BCMA)

B cell maturation antigen (BCMA, CD269, TNFRSF17) is a member of the tumor necrosis factor (TNF) receptor superfamily [24]. Because BCMA expression is restricted to (malignant) plasma cells and a subset of mature B cells, it is an attractive target for anti-MM immunotherapy [25]. Furthermore, BCMA expression persists through disease relapses and is also expressed in extramedullary plasmacytomas. BCMA ligands, B cell activating factor (BAFF), and A Proliferation Inducing Ligand (APRIL) are required for the survival of long-lived plasma cells [26]. Serum BCMA, BAFF, and APRIL are detected at increased levels in the serum of patients with MM [27]. Membrane-bound BCMA is shed as soluble BCMA (sBCMA) by a gamma secretase and can act as a decoy for BCMA-directed therapies.

### 3.2. CD74

CD74 was originally identified as the invariant chain that is associated with the α and β chains of HLA-DR (MHC class II) found on B cells and monocytes; interestingly, more than 80% of myeloma cells expresses this antigen [28]. It is involved in the formation and transport of MHC class II peptide complexes for the generation of CD4+ T cell responses and B cell maturation. Upon binding CD74 on the cell surface, anti-CD74 monoclonal antibodies become rapidly internalized, pointing to the strong potential of targeting the CD74 antigen for cancer therapy.

### 3.3. CD46

CD46 is a multifunctional protein that has a role in complement inhibition, and is expressed at a low level in normal tissue outside the placenta and prostate [23] whereas it has been shown to be highly expressed in myeloma cell lines and MM patients’ samples. The overall CD46 expression on normal hematopoietic cells is low. Notably, monocytes and granulocytes expressed relatively higher levels of CD46 compared with other normal cell populations. Interestingly, benign plasma cells from normal donors also have low CD46 antigen density, suggesting that high CD46 on MM cells occurs with malignant transition. In fact, CD46 expression was high on MM cells and interestingly, its expression is further amplified in patients with amp1q21. The *CD46* gene is located on the long arm of chromosome 1 (1q32.2) and amplification of 1q21 (amp1q21) is considered a high-risk feature that becomes more frequent at relapse.

### 3.4. SLAMF7/CS1

Signaling Lymphocyte Activation Marker Family member 7 (SLAMF7 or CS1) is a cell surface glycoprotein highly expressed on normal NK cells and plasma cells where it activates NK cell function and promotes normal B cell development [29]. Additionally, myeloma plasma cells showed high expression levels of this receptor [30], making SLAMF7 an interesting therapeutic target for multiple myeloma patients.

Elotuzumab is the first humanized IgG1 monoclonal antibody targeting SLAMF7 approved for the treatment of relapsed/refractory MM patients The phase III clinical trial Eloquent-2 demonstrated that the addiction of Elotuzumab to Lenalidomide and Dexamethasone (Elo-RD) significantly improved progression free survival (PFS) and overall survival (OS) with respect to Lenalidomide and Dexamethasone alone [31,32]. A similar clinical benefit was also observed in heavily treated patients. As a matter of fact, in the phase II Eloquent-3 trial, Elotuzumab in combination with Pomalidomide and Dexamethasone (Elo-PD) showed significantly better PFS as compared to Pomalidomide and Dexamethasone alone [33].

## 4. ADC(s) in Clinical Trials

### 4.1. Anti BCMA ADC(s)

Most ADC(s) at an advanced stage of development target BCMA. Within this group, the differences from each other are referred to the payload and linkers, which account for different efficacy and safety profiles. The most recent data concerning this group of products are summarized below:

#### 4.1.1. Belantamab Mafodotin (Belamaf)

Belantamab mafodotin (GSK2587916, Belamaf) is a BCMA directed humanized afucosylated monoclonal antibody (J6MO) conjugated via a non-cleavable linker to monomethyl auristatin-F (mcMMAF) [12]. Belamaf showed deep and pleiotropic anti-MM activity in vitro and in vivo models, without significant off-target cytotoxicity on BCMA-negative immune effector cells or bone marrow stromal cells (BMSC) [12]. The MMAF payload induces anti-proliferative and pro-apoptotic anti-MM effects. In addition, Belamaf triggers Fc-receptor mediated effector functions, including NK cell-mediated ADCC and macrophage mediated ADCP via its afucosylated Fc tail. Furthermore, Belamaf induces immunogenic cell death, a process whereby dying cancer cells elicit a host immune response by the release of neo-antigens, inducing anti-tumor immunity through immune effector cells and maturation of dendritic cells as well as recruiting macrophages for antibody dependent cellular phagocytosis. Finally, NF-kB signaling is inhibited through competition with APRIL and BAFF for binding to BCMA [12].

Based on these features, the ADC(s) was evaluated in a first-in-human, phase 1 dose-escalation/expansion study (DREAMM-1) [34,35] in an international, open label, multicenter trial for patients with relapsed/refractory multiple myeloma (RRMM), ECOG performance status (PS) 0/1, at least a previous therapy with an alkylator, PI and IMiD, and refractory to last line of treatment. Following dose escalation, the recommended dose was defined at 3.5 mg/kg every 21 days. In the extension phase, 35 patients were recruited (46–75 years, median 60); 57% of them had received 5 or more prior lines of therapy (range 1 to 10). All 35 patients had received prior PI and IMID, 89% were double refractory to a PI and an IMiD, and 37% were refractory to daratumumab. The overall response rate (ORR) was 60% (6% PR; 40% VGPR; 9% CR; 6% sCR) and the PFS was 7.9 months (median follow up 6.6 months). The most common adverse events (AEs) associated with Belamaf were corneal events (dry eye, blurring of vision, photophobia), which occurred in 22 (63%) of 35 patients in part 2, with dose interruptions/delays in 49%. In the recently reported updated analysis with an additional 14 months follow up, the ORR was confirmed (60% ORR) and the median PFS was longer at 12 months, with a median duration of response of 14.3 months [35]. In patients refractory to PI, IMiDs and Daratumumab PFS was 6.2 months. No new adverse events were reported. Based on these results, Belamaf was granted breakthrough therapy designation by the FDA, and PRIME designation by the EMA in 2017.

The phase 2 DREAMM-2 trial (NCT03525678) randomized 1:1 two different doses (2.5 mg/kg and 3.4 mg/kg) of Belamaf monotherapy for patients with RRMM [36]. Eligible patients had PS 0 to 2, with disease progression after at least three prior lines. Of note, at variance with DREAMM-1, inclusion criteria required that patients were refractory to PI, IMiD, and refractory or intolerant to an anti-CD38 mAb [36]. Due to the number of dose reductions observed with 3.4 mg/kg for toxic events, the 2.5 mg/kg dose was chosen for being further investigated. Patients were risk stratified by number of prior lines and cytogenetics. The median number of prior lines was 6–7, with 83–84% having ≥4. 196 patients were enrolled across both doses. The ORR was 31% in the 2.5 mg/kg group (with 19% achieving a VGPR or better), and 34% in the 3.4 mg/kg dose (20% VGPR or better). This response rate is lower than that observed in DREAMM-1, and it may be related to a more advanced/refractory group of patients being treated in DREAMM-2 trial. After a median follow up of 6.3 months, the median PFS was 2.9 months in the 2.5 mg/kg cohort and in the 3.4 mg/kg cohort after a median follow up of 6.9 months, the median PFS was 4.9 months. The safety profile was very similar to DREAMM-1, although grade 3 and 4 ocular events were slightly higher (27% in the 2.5 mg/kg cohort, and 21% in the 3.4 mg/kg cohort), keratopathy being the most common cause for treatment discontinuation (8% and 10% of cases). As for DREAMM1, the other most common AE reported with Belamaf in DREAMM-2 was thrombocytopenia in 35% (2.5 mg/kg) and 59% (3.4 mg/kg group), although ≥ grade 2 bleeding events occurred in 5% (2.5 mg/kg) and 17% (3.4 mg/kg cohort). Among patients with infusion-related reactions, events were predominantly grade 1–2 [36]. Given these promising results, US Food and Drug Administration (FDA) on 5 August 2020 and European Medicines Agency (EMA) on 25 August 2020 granted accelerated approval to Belamaf monotherapy treatment for adult patients with relapsed/refractory multiple myeloma who have received at least four prior therapies including an anti-CD38 monoclonal antibody, a PI, and an IMIDs.

Taking into account the significant efficacy of Belamaf in heavily pretreated patients, it can be assumed that treatment of patients who received less previous lines of therapy and are not refractory to PI and IMIDs can produce even more better results. Furthermore, the introduction of daratumumab combinations regimens as first line therapy for both transplant eligible and transplant ineligible patients paves the way for novel immunotherapeutic agents targeting different antigens aside CD38 for patients in the first relapse.

Consequently, there are a number of ongoing recruiting clinical trials of Belamaf in combination with standard and novel treatments, such as Belamaf + pomalidomide + dexamethasone (DREAMM-3, NCT03715478) [37]; Belamaf + pembrolizumab (antiPD-L1) (DREAMM-4, NCT03848845); Belamaf + dexamethasone + lenalidomide (arm A); or bortezomib (arm B) (DREAMM-6, NCT03544281) [13]. Another phase 1/2 trial (DREAMM-5, NCT04126200) was planned to explore the synergistic effects combining Belamaf with other novel anti-cancer agents, such as the T cell activating checkpoint mAbs: GSK3359609 (an IgG4 inducible T cell co-stimulatory agonist antibody that is Fc optimized to selectively enhance T cell function to enable anti-tumor responses), GSK3174998 (a humanized wild-type IgG1 anti-OX40 agonistic mAb), and PF-03084014 (a γ-secretase inhibitor) [38]. Recently, two phase 3 trial started recruiting to compare the efficacy and safety of Belamaf to currently approved standard of care therapy for relapsed refractory MM patients, in detail to daratumumab in the combination regimen with bortezomib and dexamethasone (DREAMM-7, NCT04246047) and to bortezomib in the combination regimen with pomalidomide and dexamethasone (DREAMM-8, NCT04484623). Finally, Belamaf in combination with bortezomib, lenalidomide, and dexamethasone (VRd) versus VRd alone in transplant ineligible newly diagnosed MM patients is currently being evaluating in a phase III trial (DREAMM-9, NCT04091126).

#### 4.1.2. AMG224

AMG224 is a BCMA-targeted antibody conjugated with an antitubulin maytainsinoid (mertansine, DM1) via a non-cleavable linker. The phase I, open label, first in human study enrolled 42 patients (40 receiving treatment) with RRMM and a median of 7 prior lines (range 2–11) including a PI and IMiD (NCT02561962) [39]. After dose escalation, 3 mg/kg IV three weekly was selected for dose expansion. The ORR was 23%. Common ≥ grade 3 AEs were thrombocytopenia and anemia (1 case each). Treatment-emergent ocular AEs (all grade 1 or 2) occurred in four (36%) patients in the dose expansion and included dry eye (18%), increased lacrimation (18%), and ocular hyperemia compared with Belamaf, which uses monomethyl auristatin F.

#### 4.1.3. MEDI2228 (M2)

MEDI2228 is a fully human antibody conjugated to a pyrrolobenzodiazepine (PBD) dimer via a protease cleavable linker, that is preferentially bound to membrane-bound vs. soluble BCMA, thereby more efficiently delivering the payload to MM cells [40]. After cell surface binding to BCMA, MEDI2228 is internalized and cleaved in the lysosomal compartment, releasing the active PBD dimers, a class of DNA minor groove interstrand cross-linking (ICL) agents, which cross-link DNA and lead to apoptotic cell death. MEDI2288 may provide advantages in targeting low expressing antigens and the dormant minute tumor-initiating cell populations, even at low drug-antibody ratios of ADC(s). In pre-clinical data, this agent targeted both MM cells and non-proliferating MM progenitors. Moreover, the combination of this ADC(s) with bortezomib or anti-CD38 monoclonal antibody in vivo significantly enhances efficacy to eradicate tumors [41]. MEDI2228 as monotherapy is under evaluation in a phase 1, first-in-human, open-label, dose-escalation and expansion trial (NCT03489525) recruiting RRMM patients progressed after treatment with PIs, IMiDs, and monoclonal antibodies, who are either transplant ineligible or post autologous stem cell transplant. The results were recently reported at the 62nd ASH meeting [42]. As of 15 May 2020, 82 patients treated with 2–11 lines of prior regimens received MEDI2228 during dose escalation and expansion phases. The maximum tolerated dose was 0.14 mg/kg Q3W, where more frequent treatment-related adverse events (TEAEs) were photophobia (53.7%), thrombocytopenia (31.7%), rash (29.3%), increased gamma-glutamyltransferase (24.4%), dry eye (19.5%), and pleural effusion (19.5%). No reports of keratopathy or visual acuity loss were observed. ORR was 61.0%, VGPR 24.4% (four of these patients reached immunofixation negativity) and PR 36.6%. Median duration of response was not reached. MEDI2228 pharmacokinetics was minimally impacted by circulating levels of soluble BCMA at baseline. Thirty-six patients discontinued treatment, mainly due to adverse events or progressive disease.

#### 4.1.4. CC-99712

A phase 1 first in human study (NCT04036461) is enrolling patients with RRMM treated with CC-99712 as monotherapy. The dose escalation part (Part A) of the study will evaluate the safety and tolerability of escalating doses of CC-99712, administered intravenously (IV), to determine the maximum tolerated dose (MTD) and non-tolerated dose (NTD). The expansion part (Part B) will further evaluate the safety and efficacy of CC-99712. The payload attached to this anti-BCMA mAb is currently undisclosed.

#### 4.1.5. HDP-101

HDP-101 is an anti-BCMA mAb conjugated to amanitin via a non-cleavable linker. Preclinical study discovered that when administrated at pico- to nanomolar concentrations, HDP-101 exhibited profound cytotoxicity to BCMA^+^ myeloma cell lines and non-proliferating primary MM cells isolated from patients with RRMM irrespective of BCMA expression level. Dose-dependent tumor regression after HDP-101 treatment was also observed in mouse xenograft models with both subcutaneous and systemic MM. 17p deleted MM cells appeared to be particularly sensitive [43]. In vivo study also showed a favorable safety profile in non-human primates that mainly consisted of transient, mild to moderate increase in liver enzymes, and lactate dehydrogenase. HDP-101 clinical investigation in human is going to start soon.

### 4.2. ADC(s) Addressing Different Target from BCMA

#### 4.2.1. STRO-001

STRO-001 is an ADC composed of an fully human anti-CD74, aglycosylated IgG1 SP7219 antibody conjugated to a non-cleavable linker-maytansinoid payload [22]. This compound has shown anti-myeloma effect in MM cells lines and MM xenograft models. The phase I dose escalation study (NCT03424603) in patients with MM and B cell malignancies reported two dose limiting toxicities of thromboembolic events in the first 14 patients [44], and the protocol was amended to include antithrombotic prophylaxis. The majority of AEs were grade 1/2 and no ocular toxicity has been reported. STRO-001 was granted orphan drug status for myeloma by the FDA in October 2018.

#### 4.2.2. FOR46

FOR46 consists of the CD46-specific mAb 23AG2 attached to the effector moiety MMAF via a protease-cleavable linker [23]. CD46 expression levels correlated positively with the activity of FOR46. Interestingly, because CD46 levels were upregulated on MM cells that were cocultured with BMSCs, FOR46 showed increased killing ability in the presence of BMSCs. Moreover, MM cells from patients with amplification of 1q21 had higher levels of CD46 when compared with samples from patients without this genetic abnormality. This resulted in higher activity of FOR46 in samples from patients with gain of chromosome 1q21, indicating that this ADC may be of particular interest for this high-risk patient group. In mouse models, FOR46 reduced tumor burden in a dose-dependent fashion, and improved survival compared with control mice. This compound is currently being evaluated in a phase I trial (NCT03650491) for MM patients who are refractory or intolerant to standard therapy, and who have received prior therapy with a PI, an IMiD and an anti-CD38 mAb.

#### 4.2.3. ABBV-838

ABBV-838 is an antibody-drug conjugate targeting a unique epitope of CD2 subset 1(SLAMF7/CS1), a cell-surface glycoprotein expressed on multiple myeloma cells [45]. A phase I/Ib first-in-human, dose-escalation study (NCT02462525) evaluated the safety, pharmacokinetics, and preliminary activity of ABBV-838 in patients with RRMM. As of March 2017, 75 patients received at least one dose of ABBV-838 intravenously starting from 0.6 mg/kg up to 6.0 mg/kg for 3-week dosing intervals (Q3W), up to 24 months. Assessment of alternate dosing intervals (Q1W and Q2W) was conducted in parallel. Grade 3/4/5 TEAEs were reported in 73.3% of patients across all treatment groups; most common were neutropenia (20.0%), anemia (18.7%), and leukopenia (13.3%). Grade 3/4/5 ABBV-838-related TEAEs were reported by 40.0% of patients across all treatment groups. Overall, 4.0% of patients experienced TEAEs leading to death, none ABBV-838 related. The MTD was not reached and the selected recommended dose for the expansion cohort was 5.0 mg/kg Q3W. Pharmacokinetic analysis showed that exposure was approximately dose proportional. The ORR was 10.7%; VGPR and PR were achieved by 2 (2.7%) and 6 (8.0%) patients, respectively. ABBV-838 was safe and well-tolerated in patients with RRMM, but showed a very limited efficacy.

## 5. Bispecific Monoclonal Antibodies: Design and Mechanism of Action

Bispecific T cell redirecting antibodies (BsAbs) are engineered to bind specific and selected tumor-associated antigens and the CD3 component of the T cell receptor (TCR), resulting in the immune-synapsis formation, T cell activation, and ultimately in T cell mediated killing of the neoplastic cell [46,47]. There are several advantages of this type of immunotherapeutic approach, primarily a Mayor Histocompatibility Complex (MHC) independent T cell activation, since T cell activation is independent of antigen presentation [48]. Moreover, BsAbs retain the ability to activate T cells without co-stimulatory signals by APC cells through TCR-CD3 clustering, also reducing the risk of become anergic due to TCR stimulation in the absence of costimulatory signals [47,49]. Finally, BsAbs can exert their cytotoxic properties even in the presence of low tumor antigens levels, a common occurrence in malignant cells evading immune-surveillance [50] (Figure 3).

Depending on their structure, two main subsets of BsAb can be recognized, namely BsAbs consisting only of a fragment antigen-binding (Fab) variable regions and linkers domain and BsAb, including a fragment crystallizable (Fc) domain. In addition, BsAbs that include a Fc region can further be classified among those with an IgG antibody resembling structure and those containing additional binding sites that improve the target recognition [51,52] (Figure 4).

Bs Abs lacking Fc region are generally constituted by antigen binding sites of two antibodies, generally the variable regions of the heavy and light chains (single chain variable fragment design, scFv) connected by a linker domain.

The scFv plus linker construct is known as bispecific T-cell engager molecule (BiTe), whose prototype is Blinatumomab [51,53].

The relatively simplicity and small size of BiTes is counterbalanced by several limitations in terms of short half-life and requiring administration of repetitive doses and/or continuous infusion [51,53]. Otherwise, the presence of a Fc domain critically affects the BsAbs mechanism of action since they can exhibit Fc-mediated effector functions like antibody-dependent cell mediated cytotoxicity (ADCC), antibody-dependent cell phagocytosis (ADCP), complement fixation and activation, and can ultimately increase its half-life [51,54].

BsAbs therapy shares several similarities with another popular type of immunotherapeutic approach like Chimeric Antigen Receptor expressing T-cell (CAR-T) like a MHC-independent T-cell activation and killing of tumor cells by release of granzyme B and perforin. One of the major advantage of BsAbs therapy is the “off-the shelf” availability while the main limitation is the requirement of repetitive infusion to maintain efficacy [46]. In addition, some peculiar toxicities like Cytokine Release Syndrome (CRS), neurotoxicity, and off tumor toxicities are in common [55].

CRS is a systemic inflammatory syndrome resulting from T cell activation induced by BsAbs. Excessive T-cell activation is responsible for high inflammatory cytokine levels production like Interleukin-6 (IL-6) and Interferon-α. Fever is the main symptom and in some cases, the syndrome can progress to vasodilatory shock and a capillary leak syndrome [55,56]. The etiology of neurotoxicity is unknown and is not clear if the neurotoxicity is related to the systemic CRS even though anti-IL6 tocilizumab administration was not effective in controlling the symptoms [55,57].

The ideal target of immunotherapeutic agents should be highly expressed on malignant cells, crucial for malignant cells survival and proliferation, and absent on the other tissue to avoid off target effects. Up to now, no single BsAbs is currently approved for the treatment of MM but several are under development, most of them targeting BCMA while others target CD38, CD138, CD19, and SLAMF7.

### 5.1. BCMA Targeting Bispecific Monoclonal Antibodies

#### 5.1.1. AMG420

AMG420 (previously known as BI 836909) is the first in class anti BCMA × CD3 human BiTe constituted by two scFVs. Preclinical data of unstimulated peripheral-blood mononuclear cells co-cultured with MM cell lines showed that AMG420 induced BCMA positive plasma cells lysis and targeted directed cytokine release by T-cell without affecting BCMA negative cells. In a Phase I study of NCT02514239 including 42 RRMM patients (median number of previous therapy was 5), AMG420 was administered intravenously for 4 weeks in a 6 weeks cycle, at the escalating dose of 0.2–800 μg/d [58]. Serious AEs (*n* = 20; 48%) included infections (*n* = 14) and polyneuropathy (*n* = 2); TRAEs included 2 grade 3 polyneuropathies and 1 grade 3 edema. CRS was the most common AE reported in 16 cases (38%) with only one case with grade 3 CRS, while infective events occurred in 14 patients (33%). There were no grade ≥ 3 CNS toxicities or anti-AMG 420 antibodies. In this study, 800 μg/day was considered to not be tolerable because of 1 instance each of grade 3 cytokine release syndrome and grade 3 polyneuropathy. ORR was 31%, but at the maximum tolerated dose of 400 µg/die (*n* = 10) response rate increased to 70% with 5 patients (50%) achieving MRD negativity. Median duration of response was 8.4 months.

#### 5.1.2. AMG701

AMG701is anti BCMA × CD3 human BiTe molecule binding BCMA on MM cells and CD3 on T cells, comprising two scFVs and an Fc to extend the short half-life (plasma half-life of 112 h) of AMG420 [59]. In vitro studies showed that AMG701 was able to overcame MM cell resistance to bortezomib and lenalidomide and to induce T cell mediated lysis of BCMA-positive plasma cells. Most importantly, the presence of an MM supporting microenvironment, including osteoclasts or bone marrow stromal cells, was not able to impair AMG701 lysis of MM cells. Moreover, AMG701 induced higher proliferation of T CD8+ cells and differentiation of naïve CD4+ and CD8+ T cells in effector memory and central memory cells, with these alterations being observed even after treatment. Finally, proliferating T cells treated with AMG701 were able to rapidly lyse MM cells with low BCMA expression level. To evaluate its safety and efficacy, AMG 701 was investigated in a phase 1 study (NCT03287908) of RRMM patients treated with ≥3 lines, including PIs, IMiD, and anti-CD38 mAb. Patients received AMG 701 IV escalated infusions weekly in four-week cycles until disease progression [60]. As of 2 July 2020, 75 patients received AMG 701. Patients had a median (range) of 6 (1–25) prior lines of therapy; 68% were triple refractory to a PI, an IMiD, and an anti-CD38 Ab. The most common hematological AEs were anemia (43%), neutropenia (23%), and thrombocytopenia (20%). The most common non-hematological AEs were CRS (61%), diarrhea (31%), fatigue (25%), and fever (25%). CRS was mostly grade 1 or 2. Serious AEs (*n* = 29, 39%) included infections (13), CRS (7), There were four deaths from AEs, none related to AMG 701. Reversible treatment-related neurotoxicity was seen in six patients, with earlier dose escalation with 9 mg, the response rate was 83% (5/6, 3 PRs, 2 VGPRs), with 4/5 responders being triple refractory. Across the study, responses included 4 stringent CRs (3 MRD-negative, 1 not yet tested), 1 MRD-negative CR, 6 VGPRs, and 6 PRs. Median response duration was 3.8 months, with maximum duration of 23 months; AMG 701 exposures increasing in a dose-related manner. Patient baseline sBCMA levels were identified as a determinant of AMG 701 free drug exposures; at higher doses, encouraging preliminary responses were seen even at the higher end of baseline sBCMA values.

#### 5.1.3. CC-93269

CC-93269 (formerly EM-801) is an anti BCMA × CD3 trivalent bispecific antibody in a 2+1 format with a bivalent anti-BCMA arm to increase avidity, a single anti CD3e arm and an IgG1 based Fc region to prolong half-life, with a weekly intravenous administration [61]. In vitro studies demonstrated that CC-93269 treatment resulted in a strong T cells-plasma cells interaction with increased T cell activation and function. In MM bone marrow samples, CC-93269 led to T cell activation and dose-dependent secretion of cytokines and cytolytic proteins, ultimately inducing MM cells death even in samples of heavily treated patients. In vivo tumor activity was also studied in human MM xenografted immunodeficient mice and in cynomolgus monkeys, confirming the ability to reduce BCMA positive plasma cells, even if highly proliferating [61]. The first results of a phase I dose-finding study of this agent were reported by Costa et al. [62]. Nineteen patients affected by R/R MM who received at least three previous lines of therapy (median number 6) were included in the study. Most patients received treatment with bortezomib, carfilzomib, lenalidomide, pomalidomide, and daratumumab. Of the 12 patients who received at least 6 mg, 10 patients reached at least a partial response, with 7 patients with a VGPR or better and 9 patients achieving MRD negativity. Grade 3–4 treatment related AE occurred in 15/19 cases (78.9%), mostly represented by neutropenia (52.6%), anemia (42.1%), and infections (26.3%) while approximately 90% of patients experienced CRS, mostly grade 1–2.

#### 5.1.4. Teclistamab

Teclistamab (formerly JNJ-64007957) is a humanized IgG-4 bases BCMA × CD3 bispecific antibody that induces T cell-mediated cytotoxicity against BCMA-expressing MM cells [63]. Updated results and newly available data for subcutaneous (sc) administration of an ongoing phase ½ study of teclistamab in RRMM (NCT03145181) were presented at ASH Meeting 2020 [64]. As of 20 July 2020, iv teclistamab (0.3–720 µg/kg) and sc teclistamab (80–3000 µg/kg) were received by 84 and 44 pts, respectively. Median number of prior lines of therapies was 6; 95%/79% of patients were triple-class exposed/refractory, 70%/38% penta-drug exposed/refractory, and 91% refractory to last treatment. AEs in >20% of pts (both iv and sc combined) included anemia (55%), neutropenia (55%), thrombocytopenia (41%), and leukopenia (26%), as well as non-hematologic events of CRS (53%), pyrexia (28%), diarrhea (24%), cough (23%), fatigue (23%), nausea (22%), back pain (20%), and headache (20%). 39% of patients had treatment-related grade ≥3 AEs; neutropenia (23%) and anemia (9%) were most frequent. CRS occurred in 55% and 50% of patients with iv and sc dosing, respectively, tending to occur later with sc administration. CRS events were all grade 1 (*n* = 51) or 2 (*n* = 17) and generally confined to initial doses. 5% of pts (all iv) had neurotoxicity (2% grade ≥3). Twelve patients had treatment-related infusion/injection related reaction [all i.v., 5%] and 11 injection related reactions [all sc, 25%], all grade 1/2. Grade 3 or higher infection-related AEs were reported in 15% of patients. Four gr 5 AEs were reported (all iv and considered unrelated to treatment, except for 1 case of pneumonia). Pharmacokinetic results showed that the half-life of teclistamab supports weekly iv dosing. 120 pts were evaluable for response, with the highest and most active dose levels of 270 µg/kg and 720 µg/kg weekly for iv and 720 µg/kg and 1500 µg/kg weekly for sc, respectively. Combining these iv and sc dose levels, ORR was 30/47 (63.8%, including *n* = 24 with VGPR or better and *n* = 9 with CR or better). Among 48 patients with responses across all iv and sc cohorts, median duration of response has not been reached. Of MRD-evaluable patient who had a CR, 4/5 treated in the iv cohorts and 2/2 in the sc cohorts are MRD negative at 10–6. A phase 2 monotherapy with teclistamab in RRMM (at 1500 µg/kg sc) trial is planned.

#### 5.1.5. REGN5458

REGN5458 is another BCMA × CD3 bispecific antibody that binds to both BCMA and CD3, thereby targeting MM cells with T-cell effector function via BCMA. Updated safety and response durability in a Phase 1 trial of REGN5458 monotherapy in patients with RRMM were recently reported at the 2020 ASH meeting. (NCT03761108) [65]. Enrolled patients must have progressive MM after ≥3 prior lines of systemic therapy, including a PIs, IMIDs, and an anti-CD38 Ab. Treatment consists of weekly doses of REGN5458, followed by a maintenance phase administered every 2 weeks. As of 15 June 2020, 45 patients were treated with REGN5458. Patients had a median of 5.0 prior lines of systemic therapy; 32 patients (71.1%) received a prior autologous stem cell transplant. All patients were refractory to an anti-CD38 antibody; 6.7% were triple-refractory, 33.3% were quad-refractory, and 53.3% were penta-refractory. REGN5458 was escalated in cohorts from 3−96 mg over six dose levels. The most common TRAEs included CRS (37.8%), fatigue (17.8%), nausea (17.8%), and myalgias (13.3%). CRS occurred primarily during the initial doses and was grade 1 in 88.2% of patients. No patients had grade >3 CRS. Infusion-related reactions occurred in 6.7% of patients and infection-related AEs in 46.7% of patients (grade ≥3 20%). One patient experienced a grade >3 treatment-related neurological event. Grade >3 TRAEs occurred in 28.9% of patients, with the most common being anemia (8.9%) and lymphopenia (6.7%). Serious TRAEs occurred in 22.2% of patients, with the most common due to CRS (11.1%). Grade 5 AEs (all unrelated to study drug) occurred in three patients: two sepsis and one COVID-19. ORR was 35.6% across all dose levels (60% in highest dose level), with 81.3% of responders achieving at least a VGPR, while 31.3% had a CR or stringent CR. Duration of response was >4 and >8 months in 43.8% and 18.8% of responders, respectively. The ORR in patients with extramedullary plasmacytomas was low (16.7%). Enrollment in the phase 1 dose escalation portion is ongoing, and the phase 2 portion of the study is recruiting.

#### 5.1.6. TNB-383B

TNB-383B is a BCMA × CD3 bispecific T-cell redirecting antibody incorporating a unique anti-CD3 moiety that preferentially activates effector over regulatory T-cells and uncouples cytokine release from anti-tumor activity, as well as 2 heavy-chain-only anti-BCMA moieties for a 2:1 tumor associated antigen to CD3 stoichiometry. Preliminary results from the ongoing phase 1 dose escalation and expansion first-in-human study of TNB-383B are available (NCT03933735) [66]. RRMM have been exposed to at least 3 prior lines of therapy including a PI, an INIDs and an anti-CD38 monoclonal antibody. Patients have been treated with escalating doses of TNB-383B infused IV over 1–2 h Q3W (without step-up dosing). As of 13 July 2020, 38 subjects have been dosed with TNB-383B (0.025–40 mg). The most common grade 3/4 AEs were anemia (16%) and thrombocytopenia (13%). The most common drug-related AEs were CRS (21%) and headache (13%). Cases of CRS were grade 1 (5/8) or 2 (3/8) and occurred only after the first dose of TNB-383B. No infusion-related reactions were observed. Five subjects died from underlying disease during follow-up. Fifteen subjects discontinued treatment, all of them for progressive disease. Preliminary PK data support Q3W dosing of TNB-383B. An ORR of 52% (12/23) was observed at doses ≥5.4 mg. Responses were durable (up to 24 weeks) and included 6 PR, 3 VGPR, and 3 CR. Enrollment into the dose escalation arm is ongoing.

### 5.2. Bispecific Antibodies Addressing Different Targets from BCMA

#### 5.2.1. BFCR4350A

Fc receptor-homolog 5 (FcRH5) is a type I membrane protein that is expressed on B cells and plasma cells, and is found on myeloma cells with near 100% prevalence. BFCR4350A, a humanized immunoglobulin G-based T-cell-engaging bispecific antibody, targets the most membrane-proximal domain of FcRH5 on myeloma cells and CD3 on T cells, resulting in T-cell activation and killing of myeloma cells. GO39775 (NCT03275103) is an ongoing, phase I trial evaluating BFCR4350A monotherapy in patients with RRMM for whom no other effective therapy is available or feasible [67]. Patients receive BFCR4350A by IV infusion in 21-day cycles (Q3W). with a step dose and a target dose (0.15–132 mg). At cut-off (13 April 2020), 51 pts with high-risk cytogenetics had been enrolled into Arm A. Median number of prior lines of therapy was 6. Prior treatments included PIs, IMIDs, anti-CD38 mAbs, and autologous stem cell transplant. Overall, 66.7% of patients were triple-class refractory and 94.1% were refractory to their last therapy. ORR was 51.7% (15/29) and included 3 stringent CRs, 3 CRs, 4 VGPRs, and 5 PRs. At the 3.6/20 mg dose level and above, responses were observed in patients with HR cytogenetics (9/17), triple-class refractory disease (10/20), and prior exposure to anti-CD38 mAbs (11/22), CAR-Ts (2/3), or ADC(s) (2/2). At cut-off, 6/15 patients were in response for >6 months. The most common treatment-related AE was CRS (38/51 pts, 74.5%). CRS was grade 1 in 20 patients (39.2%), grade 2 in 17 pts (33.3%), and grade 3 in 1 pt (2%). CRS was most common in the first cycle (38 pts) and was uncommon or absent in subsequent cycles. Most CRS events (49/58, 84.5%) resolved within 2 days. Other treatment-related AEs were neutropenia and lymphocyte count decreased (11.8%), aspartate aminotransferase increased and platelet count decreased (9.8%). Treatment-related grade 3–4 AEs (39.2%) in ≥3 pts were lymphocyte count decreased (6 pts, 11.8%), neutropenia (5 pts, 9.8%), anemia and platelet count decreased (3 pts each, 5.9%). No treatment-related fatalities were observed. Treatment-related AEs leading to withdrawal of treatment were uncommon; MTD was not reached. BFCR4350A PK supported the Q3W dosing regimen.

#### 5.2.2. Talquetamab

G-protein-coupled receptor class 5 member D (GPRC5D) is an orphan receptor whose transcript is highly expressed on the surface of primary MM cells but has generally limited expression elsewhere, making it an attractive therapeutic target. Talquetamab (JNJ-64407564) is a first-in-class GPRC5D × CD3 bispecific antibody that binds to GPRC5D and CD3 to induce T cell-mediated killing of GPRC5D-expressing MM cells through the recruitment and activation of T cells [68]. In preclinical models, talquetamab induced cell killing of primary MM cells and inhibited tumor formation and growth in MM mouse models. It can recruit T cells and induce tumor regression in GPRC5D+ MM murine models, which coincide with T-cell infiltration at the tumor site. Initial results from an ongoing phase 1 dose escalation study of talquetamab in RRMM (NCT03399799) are available [69]. Eligible patients have progressed on or could not tolerate established previous therapies. As of 20 July 2020, 137 pts had received talquetamab; 102 by IV (0.5–180 µg/kg) and 35 by SC (5–800 µg/kg) dosing. The median number of prior therapies was 6. 85% of patients were refractory to last line of therapy, 79% triple-class refractory, 73% penta-drug exposed, and 31% penta-drug refractory. Thirteen patients (10%) had also received selinexor and 21 (15%) had prior BCMA-directed therapy. Most common grade 3–4 AEs were lymphopenia (37%), anemia (27%), and neutropenia (25%). CRS was mostly grade 1–2 except for 5 pts with grade 3 CRS (˂8% of pts with CRS) that occurred with IV dosing; only grade 1–2 CRS was seen with SC dosing. CRS was generally confined to the first cycle. Treatment-related neurotoxicity was reported in 7 (5%) patients Infections were reported in 37% of patients (8% grade 3–4). Infusion-related reactions (IV; 15%) and injection site reactions (SC; 14%) were grade 1–2 and generally occurred in cycle 1. The maximum tolerated dose has not been defined. The half-life of talquetamab supports weekly IV dosing. IV and SC dosing of talquetamab led to comparable increases in T cell activation and cytokines. Overall response rate (ORR) for IV doses of 20–180 µg/kg was 78% (and durable with median not reached in 36/46). 

The main antibody drug conjugated and bispecific antibodies in development are listed in Table 1.

## 6. Conclusions

Although MM remains an incurable disease, the paradigm treatment of patients indeed continues to evolve with new classes of agents that contribute to improvement in overall survival of patients preserving a good quality of life. Recent data on immunotherapy with CAR-T are impressive, although with a relevant toxic profile [70]; however, MM patients are mostly characterized by old age and several comorbidities, preventing their possibility to benefit from this approach.

ADC(s) [71] and BsAbs [72] are designed with the aim to deliver targeted therapy maximizing efficacy and limiting systemic toxicity. Although most advanced ADC(s) and BsAbs target BCMA as CAR-T cells, they have some advantages as compared to CAR-T (Table 2). They are an “off the shelf” product and there is no delay between the decision to treat the patient and administration of the drug. In addition, ADC(s) and Bispecific Antibodies can be readily combined with other treatments to increase efficacy.

In contrast to CAR-T for which usually only one infusion is needed, with ADC(s) and BsAbs, multiple dosing is expected to elicit a durable response; furthermore, particular for ADC(s), infusion reactions are mostly grade 1–2, and infusions is intermittent, and not weekly or continuous. Reducing/managing toxicity, however, remains the most crucial issue for ADC(s) and BsAbs. Concerning ADC(s), a particular effort is doing to look for alternative linker-payload constructs maintaining efficacy without complications.

In conclusion, the development of these new treatments for MM patients is going to greatly contribute to improving outcomes for a wide group of patients. Their use in earlier phases of the disease and the possibility of combination with other active drugs will play a relevant role in the future therapeutic landscape of these patients.

## Figures and Tables

**Figure 1 pharmaceuticals-14-00040-f001:**
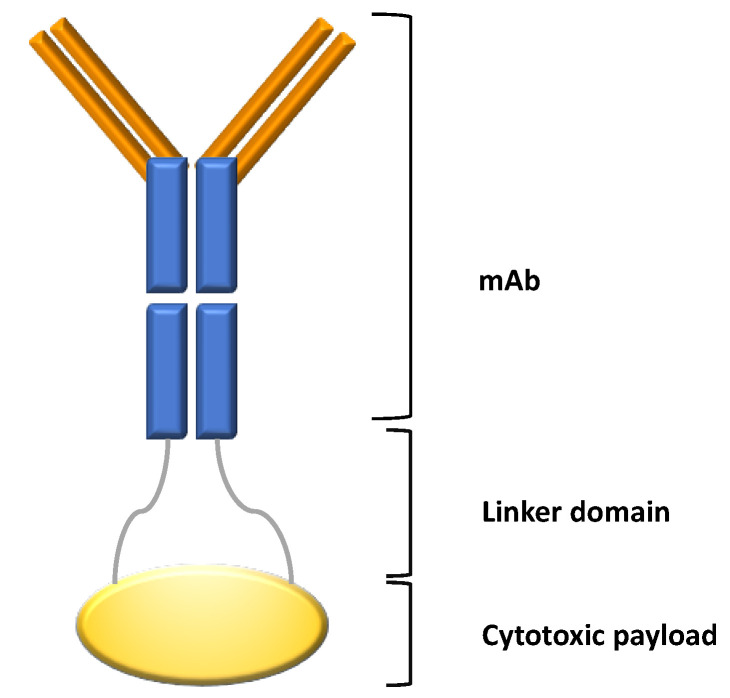
Design of antibody drug conjugate (ADC). ADC is composed of a monoclonal antibody (mAb) containing the antigen binding site connected by a linker domain to the cytotoxic drug (payload).

**Figure 2 pharmaceuticals-14-00040-f002:**
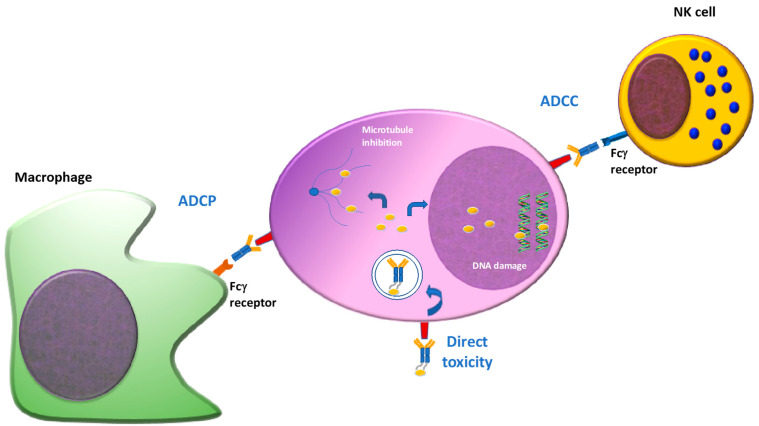
Mechanism of action of ADC(s). ADC(s) can exert their cytotoxic properties directly or indirectly using the immune system. Once the antibody binds the target antigen on the cell surface, it is internalized and the payload is released into the cytoplasm, interfering with the cellular functions (DNA damage, microtubules inhibition), inducing apoptosis. Otherwise, some ADC(s) can activate immune effector cells by Fc-Fc receptor interaction, inducing ADCC or ADCP. ADCC: antibody dependent cellular cytoxocity. ADCP: antibody dependent cellular phagocytosis.

**Figure 3 pharmaceuticals-14-00040-f003:**
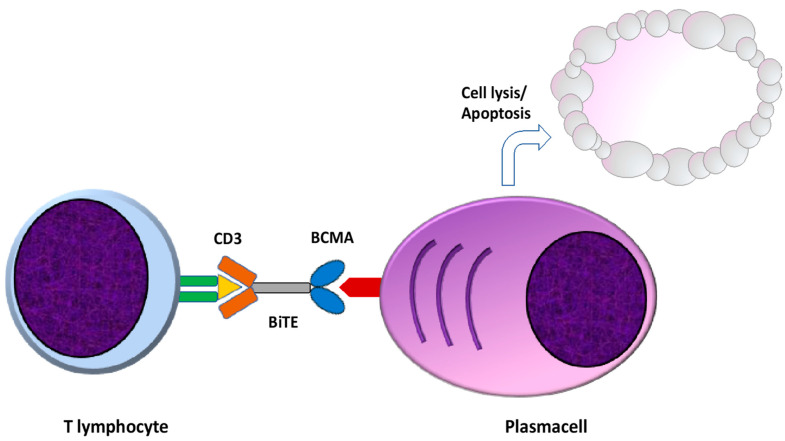
Mechanism of action of bispecific antibodies. Bispecific antibodies (BsAbs) are engineered to bind specific tumor associated antigens and the CD3 component of the TCR, resulting in immune-synapsis formation and MHC and APC independent T cell activation and T cell mediated killing of the neoplastic cell. TCR: T cell receptor. MHC: major histocompatibility complex. APC: antigen presenting cells.

**Figure 4 pharmaceuticals-14-00040-f004:**
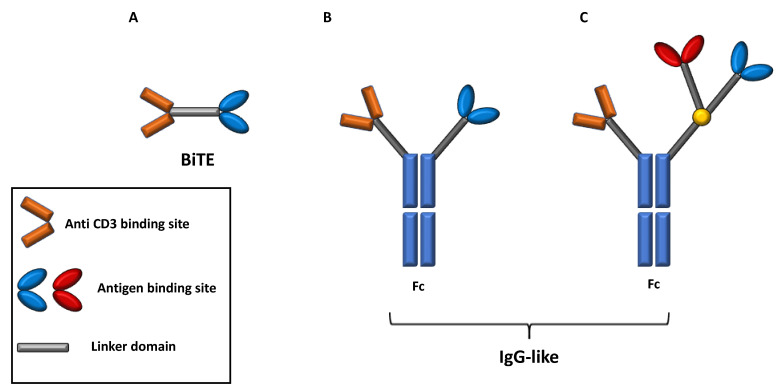
Design and subtypes of bispecific antibodies. BsAbs can be classified as BsAbs containing two Fab domain connected by a linker domain and those containing a Fc domain. BsAbs containing only Fab domains like BiTE (Panel (**A**)) are the smallest and simpler type and activate T-cells inducing TCR clustering. BsAbs containing a Fc region, instead, can exhibit antibody specific functions like ADCC and ADCP and can be further distinguished in those with IgG antibody structure and those containing additional binding sites that improve the target recognition (Panel (**B**,**C**)). BsAbs: bispecific antibodies. TCR: T cell receptor. MHC: major histocompatibility complex. ADCC: antibody dependent cellular cytoxocity. ADCP: antibody dependent cellular phagocytosis.

**Table 1 pharmaceuticals-14-00040-t001:** Main antibody drug conjugated and bispecific antibodies in development.

Name	Type	Target	References
Belantamab mafodotin	monomethyl auristatin-F ADC	BCMA	[34,35,36]
AMG224	Mertansine ADC	BCMA	[39]
MEDI2228	Pyrrolobenzodiazepine ADC	BCMA	[41,42]
CC-99712	Undisclosed ADC	BCMA	
HDP-101	Amanitin ADC	BCMA	[43]
STRO-001	Maytansinoid ADC	CD74	[44]
FOR46	monomethyl auristatin-F ADC	CD46	
ABBV-838	monomethyl auristatin-E ADC	SLAMF7/CS1	[45]
AMG420	BCMA × CD3 BiTe	BCMA	[58]
AMG701	BCMA × CD3 BiTe	BCMA	[59,60]
CC-93269	BCMA × CD3 trivalent BsAb	BCMA	[61,62]
Teclistamab	BCMA × CD3 BsAb	BCMA	[63,64]
REGN5458	BCMA × CD3 BsAb	BCMA	[65]
TNB-383B	BCMA × CD3 BiTe	BCMA	[66]
BFCR4350A	FcRH5 × CD3 BiTe	FcRH5	[67]
Talquetamab	GPRC5D × CD3 BsAb	GPRC5D	[68]

BCMA: B cell maturation antigen. SLAMF7: Signaling Lymphocyte Activation Marker Family member 7. FcRH5: Fc receptor-homolog 5. GPRC5D: G-protein-coupled receptor class 5 member D. ADC: antibodies drug conjugate. BsAb: bispecific antibody.

**Table 2 pharmaceuticals-14-00040-t002:** Comparison of immunotherapy strategies for multiple myeloma.

	Antibody-Drug Conjugate(s)	Bispecific Antibodies	CAR-T
**Pros**	“Off the shelf“ product	“Off the shelf” product	High response in the relapsed refractory setting
	Independent from host immune function	High response in the relapsed refractory setting	Only one treatment required
	No delay in administration	No delay in administration	
	Can be given in the community setting	Can be given in the community setting?	
**Cons**	High cost	High cost	High cost
	Continuous therapy	Continuous therapy	Long production time (4–6 weeks)
	Higher doses may be required for antigen downmodulation	CRS and ICANs toxicity	CRS and ICANs toxicity
	Payload mediated toxicity		Requires conditioning therapy
	Potential lower response rate		Require adequate lymphocyte count and function

CRS: Cytokine release syndrome. ICANs: immune effector cell associated neurotoxicity syndrome.

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
