# Peer review of "Drug Conjugated and Bispecific Antibodies for Multiple Myeloma: Improving Immunotherapies off the Shelf"

_pharmaceuticals, 2021, doi:10.3390/ph14010040_

Round 1
Reviewer 1 Report
Minor points should be addressed:
- For guidance of non-expert readers, company names for abbreviations (e.g. BI) should be indicated at a proper place.
- For sake of clarity, the presented antibodies should be compiled in an additional table giving a few key features.
Author Response
As suggested we added company names at proper places and an additional table with the antibodies and key features.
Reviewer 2 Report
In this review, the authors reviewed the characteristics, mechanism of action and clinical data available for most relevant ADC and BsAbs against multiple myeloma. This is a well-written, comprehensie and updated review. Only a few minor comments:
Abstract: Please provide a more focused abstract on the current landscape regarding antibody drug conjugated and bispecific antibodies (eg. basic mechanism of action, referral to the most promising agents) and not just an introductory section on the main paper.
Page 6: Please add also info on DREAMM 8 and 9 clinical trials.
Please add data regarding the FDA approval of belantamab mafodotin.
Please add a critical comment on the anticipated role of belantamab mafodotin in the continuum of treatment for patiens with multiple myeloma (only for heavily pretreated patients or at earlier lines of therapy?)
Please add updated data on MEDI2228 from ASH 2020.
Author Response
Abstract: Please provide a more focused abstract on the current landscape regarding antibody drug conjugated and bispecific antibodies (eg. basic mechanism of action, referral to the most promising agents) and not just an introductory section on the main paper.
R: we ameliorated the abstract including mechanism of action and most promising antibody drug conjugated and bispecific antibodies
Page 6: Please add also info on DREAMM 8 and 9 clinical trials.
Please add data regarding the FDA approval of belantamab mafodotin.
Please add a critical comment on the anticipated role of belantamab mafodotin in the continuum of treatment for patients with multiple myeloma (only for heavily pretreated patients or at earlier lines of therapy?)
R: as suggested we added info on DREAMM 8 and 9 clinical trials and on FDA approval of Belantamab. We also added a comment on its anticipated role in the treatment of Multiple Myeloma patients.
Please add updated data on MEDI2228 from ASH 2020.
R: we added updated ASH 2020 results of MEDI2228.
Reviewer 3 Report
In this manuscript the authors provide a comprehensive overview on antibody-drug conjugates (ADCs) and bispecific antibodies for Multiple Myeloma (MM) targeting.
In general the article is timely and well written although some minor points require revision.
- I would suggest using the more commonly used definition “antibody-drug conjugate(s)” rather than “antibody drug conjugated” in the text when mentioned for the first time, and the acronym ADC(s) throughout the manuscript.
- Line 65 the sentences: “When the antibody recognizes and binds to its specific receptor, the complex antigen-receptor is internalized within the cell. Once the antibody with conjugated payload is introduced within the cells, the complex can disrupted by linker cleavage or antibody degradation and the paylod is released and free to interfere with vital cellular functions.” are wordy and the use of the word “receptor” to indicate the ADC target antigen is confusing. I would suggest modifying the text as follows: “ When the ADC binds to its target antigen at the cell surface the antigen/ADC complex is internalized within the cell in the endo-lysosomal compartment. Here, the ADC can be disrupted by linker cleavage or antibody degradation and the payload is released and free to interfere with vital cellular functions.”
- In paragraph 3 the authors describe the antigen-targets for ADC, including BCMa, CD74, CD46. I would suggest briefly mentioning also SLAMF7/CS1, which has been also the target molecule for elotuzumab, the first mAb approved for MM therapy.
- Figure 3: It seems that the only way of myeloma cell killing by BiMab-targeted T cells is induction of apoptosis. The perforin-mediated lytic mechanism has been excluded? If not please use “Cell Lysis/apoptosis” in the figure.
Author Response
In this manuscript the authors provide a comprehensive overview on antibody-drug conjugates (ADCs) and bispecific antibodies for Multiple Myeloma (MM) targeting.
In general the article is timely and well written although some minor points require revision
I would suggest using the more commonly used definition “antibody-drug conjugate(s)” rather than “antibody drug conjugated” in the text when mentioned for the first time, and the acronym ADC(s) throughout the manuscript.
R: we used the definition of antibody-drug conjugate and the ADC acronym as recommended.
Line 65 the sentences: “When the antibody recognizes and binds to its specific receptor, the complex antigen-receptor is internalized within the cell. Once the antibody with conjugated payload is introduced within the cells, the complex can disrupted by linker cleavage or antibody degradation and the paylod is released and free to interfere with vital cellular functions.” are wordy and the use of the word “receptor” to indicate the ADC target antigen is confusing. I would suggest modifying the text as follows: “ When the ADC binds to its target antigen at the cell surface the antigen/ADC complex is internalized within the cell in the endo-lysosomal compartment. Here, the ADC can be disrupted by linker cleavage or antibody degradation and the payload is released and free to interfere with vital cellular functions.”
R: we modified the following paragraph as proposed.
In paragraph 3 the authors describe the antigen-targets for ADC, including BCMa, CD74, CD46. I would suggest briefly mentioning also SLAMF7/CS1, which has been also the target molecule for elotuzumab, the first mAb approved for MM therapy.
As suggested, we added a specific paragraph for SLAMF7 antigen.
Figure 3: It seems that the only way of myeloma cell killing by BiMab-targeted T cells is induction of apoptosis. The perforin-mediated lytic mechanism has been excluded? If not please use “Cell Lysis/ apoptosis” in the figure.
R: We modified Figure 3 as indicated.